# TCR *CDR3* Sequencing as a Clue to Elucidate the Landscape of Dysimmunity in Patients with Primary Immune Thrombocytopenia

**DOI:** 10.3390/jcm11195665

**Published:** 2022-09-26

**Authors:** Lili Ji, Yanxia Zhan, Boting Wu, Pu Chen, Luya Cheng, Yang Ke, Xibing Zhuang, Fanli Hua, Lihua Sun, Hao Chen, Feng Li, Yunfeng Cheng

**Affiliations:** 1Department of Hematology, Zhongshan Hospital, Fudan University, Shanghai 200031, China; 2Department of Transfusion Medicine, Zhongshan Hospital, Fudan University, Shanghai 200031, China; 3Department of Laboratory Medicine, Zhongshan Hospital, Fudan University, Shanghai 200032, China; 4Center for Tumor Diagnosis and Therapy, Jinshan Hospital, Fudan University, Shanghai 201508, China; 5Department of Hematology, Zhongshan Hospital Qingpu Branch, Fudan University, Shanghai 201700, China; 6Department of Thoracic Surgery, Zhongshan Hospital Xuhui Branch, Fudan University, Shanghai 200031, China; 7Institute of Clinical Science, Zhongshan Hospital, Fudan University, Shanghai 200031, China

**Keywords:** T cell receptor, CDR3, primary immune thrombocytopenia

## Abstract

**Background:** Primary immune thrombocytopenia (ITP) is an autoimmune disorder. The existence of autoreactive T cells has long been proposed in ITP. Yet the identification of autoreactive T cells has not been achieved, which is an important step to elucidate the pathogenesis of ITP. **Methods:** ITP patients’ peripheral blood was collected prior to the treatment and one month after initiating dexamethasone treatment per related therapeutic guideline. Serum cytokines were profiled to examine T cell subtypes imbalance using a protein chip. TCR Vβ analysis in CD8^+^T cells of ITP patients, and TCR CDR3 DNA sequencing of CD4^+^T and CD8^+^T cells were performed to determine the autoreactive T cells’ clones. **Results:** Cytokine profiling revealed imbalanced distribution of T cells subtypes, which was confirmed by CD4^+^T and CD8^+^T cells’ oligoclonal expansion of TCR Vβ analysis and TCR CDR3 DNA sequencing. VDJ segments were found to be more frequently presented in ITP patients, when compared with health controls. There was an individualized CD4^+^T cell or CD8+T cell CDR3 sequence in each ITP patient. **Conclusions:** The present study revealed that T cell clones expanded in ITP patients’ peripheral blood, and each clone had an individualized TCR CDR3 sequence. The expanded T cell clones preferred to use some specific VDJ segment. Further studies are warranted to get access to individualized treatment such as Car-T in patients with ITP.

## 1. Background

Primary immune thrombocytopenia (ITP) is an acquired autoimmune disorder. Numerous studies have highlighted the key role of T cells in the pathogenesis of ITP [1,2]. The Th1/Th2 imbalance was first detected in no other autoimmune diseases but ITP [3]. Treg cells, as a modulator of the immune system, is numerically and functionally insufficient in ITP patients [4,5,6,7,8]. Diagnostic investigations into immune functions and characteristics in ITP, which may potentially be exploited to help predict responses and thereby distinguish therapeutic responders from non-responders [9]. Our previous study revealed the role of Th17/Treg imbalance in the pathogenesis of ITP [10]. Besides the CD4^+^ T cells, CD8^+^ T cells also participate in increasing platelet apoptosis and clearance [11,12,13].

HLA locus [14,15] genetic predispositions have been confirmed in autoimmune diseases, including ITP [16,17,18]. Auto-reactive T cells’ existence in ITP has been reported [19,20], yet the definite autoreactive T cell clones have not been identified, which withholds subsequent studies, such as molecular mimicry theory in the pathogenesis of ITP, and novel targeted therapy for ITP.

Theoretically, the autoreactive pathogenic T cells were activated by T cell receptor (TCR) complex combining with peptide-MHC molecule ligand [21]. These T cell clones then expanded, verifying the oligo-clonal phenomenon in ITP [22,23]. And these expanded pathogenic T cells played their pathogenic role by secreting cytokines (CD4^+^T cells) or exerting cytotoxic effect (CD8^+^ T cells).

The TCR repertoire is of great diversity, which enables T cells to recognize different types of antigens. It has been reported that over one million TCRB sequences were detected in healthy donors [24]. In addition, TCR repertoire is not static, T cell clones expand when stimulated by antigen they recognizing, and contract after clearance of the antigen. So TCR repertoire is continuously molded by the input of new T cells and response to immune challenge [25]. These two characteristics have greatly hampered the identification of autoreactive T cell clones.

Over 90% TCR of human are composed of α chain and β chain. Complementary determine region 3 (CDR3), locates on the β chain, is the combining area of antigen. CDR3 domain, approximately 45 nucleotides long, comprises the VJ (for TCR-α) or VDJ (for TCR-β) junction, and represents a primary source of sequence diversification and TCR variability. TCR sequencing method mostly involves PCR amplification and sequencing of CDR3 region on TCR β chain [26]. Next generation sequencing (NGS) technology, a massively parallel sequencing technology that allows the sequencing of thousands to millions of DNA molecules simultaneously, has been used in rheumatoid arthritis (RA) to search the autoreactive T cell clones [27], in which highly expanded T cell clones in joints were found, underscoring the importance of autoreactive T cell clones detection.

The present study aimed to investigate the autoreactive T cell clones in ITP, and to profile the T cells repertoire of ITP and its variation after the treatment. Specifically the cytokines profile in ITP patients’ serum and TCR Vβ distribution of CD8^+^T cells of ITP patients’ peripheral blood were examined. T cell CDR3 region of ITP patients were also sequenced using NGS technology. Given the fact that CD4^+^ TCR and CD8^+^ TCR combining MHCII and MHCI molecules respectively, these two subtypes of T cells were separately sequenced.

## 2. Methods

From April 2016 to Jan 2018, patients with newly diagnosed ITP according to the ITP diagnosis criteria proposed by an international working group [28] were consecutively recruited. Secondary ITP and pregnant patients were excluded. All patients’ platelet counts were less than 30 × 10^9^/L, and they received 40 mg oral dexamethasone for consecutive 4 days [29]. One month after the treatment, therapeutic efficacies were validated in accordance with the Vicenza Consensus Conference [28]. Twenty mL venous blood samples were collected from each patient upon enrollment and one-month post initial treatment for analysis.

The study was approved by the Institutional Review Board of Zhongshan Hospital, Fudan University. Written informed consent was obtained from each patient prior to the enrollment.

### 2.1. Sample Preparation

Venous blood samples were collected in ethylenediaminetetraacetic acid-treated tubes. The plasma of all samples was collected after centrifugation (2200 rpm at room temperature for 10 min), and stored at −80 °C refrigerator. Peripheral blood mononuclear cells (PBMCs) were collected by Ficoll-Hypaque gradient centrifugation (blood samples were diluted 1:2 with Hanks balanced salt solution (HBSS) and centrifugated at 2200 rpm at room temperature for 15 min). Washed and re-suspended, isolated peripheral blood mononuclear cells (PBMCs) were cryopreserved in fetal bovine serum containing 10% dimethyl sulfoxide (DMSO), and stored in liquid nitrogen.

### 2.2. Plasma Protein Profiling

The plasma samples were thawed to room temperature. The protein concentration of plasma was adjusted to 50–500 μg/mL with BCA protein concentration test kit (Sigma-Aldrich, St. Louis, MI, USA). Forty cytokines (Activin A, AgRP, Angiogenin, ANG-1, Angiostatin, Cathepsin S, CD40, Cripto-1, DAN, DKK-1, E-Cadherin, TROP1, Fas Ligand, Fc γ RIIB/C, Follistatin, Galectin-7, ICAM-2, IL-13 R1, IL-13 R α2, IL-17β, IL-2 R α, IL-2 R β, IL-23, LAP/TGF β1, NrCAM, PAI-1, PDGF-αβ, Resistin, SDF-1 β, gp130, Shh-N, Siglec-5, ST2, TGF β2, Tie-2, TPO, TRAIL R4, TREM-1, VEGF-C, and VEGF R1) were tested by Human inflammatory cytokine quantitative protein chip kit (RayBiotech, Norcross, GA, USA) per manufacturer’s protocol. The fluorescence was read by Luxscan 10K dual laser microarray chip scanner (Capitalbio, Beijing, China), data were analyzed by ScanArray Express.

### 2.3. TCR Vβ Subfamily Analysis of CD8^+^T Cells by Flowcytometry

Cryopreserved PBMCs were thawed at 37 °C, washed twice with HBSS. Dead cells were removed by dead cell removal kit (Miltenyi Biotec, Bergisch Gladbach, Germany). Then, 10^10^ PBMCs were stained with anti-human CD8^+^-Pc5 antibody, anti-human CD28^+^-PE or -FITC antibodies, and isotype antibodies for flowcytometry analysis (BD Bioscience, USA, In addition, anti-human TCRVβ1, Vβ2, Vβ3, Vβ5.1, Vβ5.2, Vβ6.7, Vβ7, Vβ8, Vβ11, Vβ12, Vβ13.1, Vβ13.6, Vβ14, Vβ16, Vβ17, Vβ20, Vβ21.3, Vβ22, -FITC antibodies, anti-human TCRVβ5.3, Vβ9, Vβ18, Vβ23, -PE antibodies (Beckman-Coulter, USA) were also used as well. The analysis was performed on FACS Aria II workstation (BD Bioscience, USA).

### 2.4. Cell Purification and DNA Extraction

PBMCs were used for TCR CDR3 sequence analysis. First, CD4^+^ T cells and CD8^+^ T cells were purified by CD4^+^ microbeads and CD8^+^ microbeads, (Miltenyi Biotec, USA) respectively. The purified cells were verified by flowcytomotry (CD4 FITC, and CD8APC, eBioscience, San Diego, CA, USA). CD4^+^ T cells and CD8^+^ T cells’ DNA were respectively extracted using DNA extraction kit (Tiangen, Beijing, China) per manufacture’s protocol.

### 2.5. Construction of Genomic DNA Library and Sequence

The genomic DNA library construction and sequence were performed according to Beijing Genomics institution’s (BGI, Shenzhen, China) SOP. Briefly, suck 1 uL DNA, add it into qubit tube, add 1 × dsDNA HS working solution, adjust the system solution to 200 uL, shake it fully, incubate for 2 min, and then add sample to qubit fluorescence meter to read the DNA concentration. According to the concentration of DNA, the initial amount of DNA was 400 ng, which was added into the system containing V-region primer and J-region primer of Illumina sequencing splice sequence, and the multiple PCR reaction was carried out with Qiagen kit (Qiagen, Germantown, MD, USA). The multiplex PCR products were purified by using 1-fold volume of Agencourt AMPure XP magnetic beads (Bechman, CA, USA). The purified and recovered DNA was screened by using the Agencourt AMPure XP magnetic beads. The recovered DNA products were amplified by two rounds of primers with Illumina Flow Cell sequence (Illumina, CA, USA). The PCR products were agarose gel electrophoresis, and the target fragments were cut and purified by QIAquick Gel Extraction Kit). The products were dissolved in Elution Buffer and labeled with library tags. The TCR Region Library was sequenced by Illumina Hiseq. NaOH was added to denature the chain into single chain, and diluted to the expected on-board density. The denatured and diluted library was added into FlowCell, hybridized with the connector on FlowCell, then completed the bridge PCR amplification on the cluster generation platform cBOT, and finally conduct on-board sequencing, with the sequencing type of PE101.

The sequencing data were analyzed using MiXCR (Milaboratory, CA, USA, MiXCR THE ULTIMATE SOLUTION FOR IMMUNOGENETICS. Available online: https://mixcr.readthedocs.io/en/master/ accessed on 23 January 2019), a consistent sequence was recorded as one clone. Each clone’s frequency was divided by the sum of all clones’ frequencies as the percentage to TCR receptor library, as the number parameter of this specific clone.

### 2.6. Statistical Analysis

All analyses were performed with STATA 7.0 software (StataCorp LP, College Station, TX, USA). Data were expressed as mean ± SD. Normality was assessed by Shapiro-Wilk W test. In pairwise comparison, student *t* test and Wilcoxon rank-sum (Mann-Whitney) test were used for data fulfilled normal distribution and for those did not, respectively. When multiple groups were compared, One Way ANOVA and Kruskal Wallis test were used for data fulfilled normal distribution and for those did not, respectively. For all tests, two-sided *p* values less than 0.05 were considered statistically significant.

## 3. Results

From April 2016 to Jan 2018, 27 patients (18–77 years old, median age 47 years) with newly diagnosed ITP were enrolled. All of the patients received dexamethasone therapy as their platelet counts were less than 30 × 10^9^/L. The baseline characteristics of the patients were listed in Table 1.

### 3.1. T cells Related Cytokines Concentration Related to ITP Disease Activity

Plasma protein chip revealed that serum levels of eight cytokines were different between ITP patients and normal controls (NCs). The levels of IL-23, IL-17β, Galectin-7, LAP/TGF β1, PDGF-αβ, ANG-1, E-cadherin were decreased in ITP patients at disease onset, and increased at disease remission, or stayed low if disease didn’t remit (Figure 1A–G). AgRP level was lower in therapy-responded patients than in NCs before treatment, and was comparative to NCs after treatment. In patients who were irresponsible to the treatment, the levels of AgRP remained similar to that of NCs (Figure 1H).

### 3.2. CD8^+^T Cells’ Oligoclonal Proliferation in ITP Patients by Vβ Flow Cytometry

The proliferation of TCR Vβ oligonucleotides was observed in almost all patients, and the average number of oligonucleotides of TCR Vβ subfamily in CD8+ T cells was about 4–5 at onset. TCR Vβ flow cytometry showed CD8^+^T cells’ oligoclonal proliferation in each ITP patient, while TCR Vβ distribution was normal in healthy individuals. The types of oligoclonal TCR Vβ subfamily were not identical in different ITP patients, yet several types were commonly over-proliferative (Table 2). After treatment, although the oligoclonal proliferation of CD8^+^ T cells still existed, there was a smaller number of oligoclonal clone in remission group, with an average of 2–3 clones, while there was a slight increased number of oligoclonal in the NR group, with about 6–7 clones (Table 2).

### 3.3. CD4^+^T and CD8^+^ T Cells’ TCR CDR3 Repertoire Skewed before and after Treatment

Before treatment, ITP patients’ CD4^+^T and CD8^+^T cell repertoire showed a skewed distribution. The total number of clones of CD4 ^+^ T cells was 262400.60 ± 184130.07. Most clones (262376.00 ± 184138.74) were low-frequency ones (<0.1% of total TCR analyzed, Table 3). Few clones were of higher frequency (Table 3). The CD4 ^+^ TCR clones were divided into six grades according to their percentage of total TCR. The percentage of TCR CDR3 in each grade was shown in Figure 2A. TCR of low frequency (<0.1% grade) accounted for (81.90% ± 14.24%) of the total TCR. As these low frequent TCR clones were of small size and composed with numerous types, the study focused on the higher frequent clones. The percentage of TCR of highest frequency (>0.5%) was numerically higher than that of others, yet didn’t reach statistical significance (*p* = 0.1453, Figure 2A).

In the set of CD8^+^ T cells, the distribution was similar. The total number of clones was (244923.21 ± 69294.59), while low-frequency clones accounted (244826.60 ± 69288.66) (Table 4). The percentage of TCR CDR3 in each of the six grades was shown in Figure 2B. TCR of low frequency (<0.1% grade) accounted for (53.90% ± 21.30%) of the total TCR. The percentage of TCR of highest frequency (>0.5%) was higher than that of others (*p* = 0.0038, Figure 2B).

After the treatment of dexamethasone, ITP patients’ CD4^+^ T and CD8^+^ T cell repertoire remained a skewed distribution. Although all 5 patients obtained complete remission (CR), the sequence results showed that the skewed distribution of CD4^+^ TCR prior to the therapy still exists. After treatment, the number of TCR clones was (268533.40 ± 56210.77), similar to the base level (*p* = 0.6858). Most clones (268507.40 ± 56213.30) were low-frequency ones (<0.1% of total TCR analyzed, Table 5). The percentage of TCR CDR3 in each grade was shown in Figure 3A. The percentage of each grade didn’t differ after treatment. The percentage of high-frequent clones (>0.5%) was higher than other 4 grades (*p* = 0.0010), excluding the mini size of clones (<0.1%).

After treatment, the number of CD8^+^ TCR clones was (268427.20 ± 120730), similar to the base level (*p* = 0.2249). Most clones (268357.00 ± 120752.43) were low-frequency ones (<0.1% of total TCR analyzed, Table 6). The percentage of TCR CDR3 in each grade was shown in Figure 3B. The percentage of grade (0.1%–0.2%) was higher than that before treatment (*p* = 0.0431), and other grades didn’t differ. The percentage of high-frequent clones (>0.5%) was higher than other 4 grades (*p* = 0.0061), excluding the mini size of clones (<0.1%).

### 3.4. The High Frequent TCR Clone Remain Unchanged after Treatment

Previous results showed shewed TCR repertoire distribution of CD4^+^ T or CD8^+^ T cells, especially those higher than 0.5% (defined as highly expanded clones, HECs [27]), still existed when the disease remitted in the 5 patients. That raised a new question: whether the HECs after treatment were all the same HECs before treatment? Or, the HECs after treatment were newly expanded clones, as the HECs before treatment were eliminated by dexamethasone? So the detailed nucleotide sequence of each patient before and after treatment was inspected. The comparison found that HECs before treatment still existed as HECs after treatment in each patient. A representative HECs sequence of one patient was shown in Table 7. A diagram was draw to demonstrate the HECs comparison of the 5 patients (Figure 4).

### 3.5. CD4^+^ TCR HECs Didn’t Share Individualized CDR3 Sequence with CD8^+^ TCR HECs in Each of the Patient, Any Patient Didn’t Share a Single CDR3 Sequence of TCR with Any Other Patient

To find any possible clue of autoreactive T cells in these 5 patients, each patient’s CD4^+^ TCR HECs’ CDR3 sequence were compared with her own CD8^+^ TCR HECs. Results showed that no clone shared a same CDR3 sequence. Nor shared sequences were found between different patients.

### 3.6. VDJ Usage Tendency of TCR CDR3 in ITP Patients

VDJ usage of TCR CDR3 of these 5 patients were also analyzed. Results showed that the VDJ usage by CD4^+^ T cells remained similar after treatment and so did CD8^+^ T cells. CD4^+^ T cells were more likely to use TRBV12-4, TRBV6-4, TRBV25-1, TRBV6-7, TRBV28, TRBV21-1, and TRBJ1.1. CD8^+^ T cells preferred TRBV12-4, TRBV6-4, TRBV25-1, TRBV6-7, TRBV28, TRBV9, TRBV2, TRBJ1.1, TRBJ1.6, and TRBJ2.2.

## 4. Discussion

ITP is an autoimmune disease, in which auto-reactive CD4^+^ T cells are thought to play pivotal roles in helping B cells to produce auto-reactive antibodies. More recently, BTKis have been under development for immune-mediated diseases, such as ITP [30]. Besides, auto-reactive CD8^+^ T cells contribute to disease pathogenesis by attacking platelet and megakaryocyte directly. The auto-reactive T cells’ existence in ITP has been confirmed [19,20]. There have been numerous attempts to identify the “criminal” T cells, which could make precise individual treatment possible, but unfortunately failed in proving T cell oligoclonal proliferation [22,23].

The current study found that the levels of several T cells’ cytokines were correlated with the disease activity. IL-23 and IL-17 decreased in active ITP patients, and recovered to normal level when disease remitted. Adams et al. [31] has confirmed Th17/Treg cells imbalance in ITP patients [10], which seemed to be contradictory to current results. The different timepoint of cytokine sampling in the course of the disease might account for this. The cytokines were examined on enrollment and at one month after initial treatment in this study. The time interval might to be too short to see the decline of IL-23/IL-17 after inital treatment. Interestingly, another cytokine, AgRP, decreased in active ITP patients who would response to dexamethasone, and recovered to normal level after treatment. In NR patients, AgRP stayed at normal level before treatment. As we know, AgRP is a protein related to systemic energy metabolism [32]. We previously showed a paradox in Treg cells’ IL-10 production and secretion [6], which might be related to cell metabolism (data not publicized). It warrants new inspection to confirm whether AgRP changed by abnormal cell metabolism. In any case, the data of serum cytokine implied that it was not advisable to infer T cell abnormalities from the perspective of cytokine changes.

Flow cytometry analysis and sequence technology both confirmed the T cells oligoclonality. TCR repertoire has been found related to a wide range of diseases, including malignancy, autoimmune disorders and infectious diseases. Characterizing TCR repertoires is a priority of great scientific interest and potential clinical utility [26]. More than ten years ago, it has been proposed the possibility to make TCR vaccination to treat autoimmune disease [33]. And on the way to find the criminal T cell clones, some inspiring results have been obtained in some diseases [34,35].

TCR CDR3 sequencing is the most common method of TCR repertoire study. It has been used in autoimmune disease studies to find auto-reactive T cells [27,36], and to explain the “molecular mimicry theory”, one of the pathogenesis of autoimmune disease [37]. In the present study, newly diagnosed ITP patients were enrolled, their CDR3 were sequenced with NGS technology for CD4^+^ T cells and CD8^+^ T cells separately. In addition, the CDR3 sequence after standard treatment was examined to study the impact of the treatment on TCR repertoire.

Our data showed that CD4^+^ T and CD8^+^ T cells had over 2 million clones, before and after treatment, confirming the diversity of TCR repertoires. The data was comparative with former report [24], indicating that the diversity of TCR repertoire in newly diagnosed ITP patients is not destroyed by the disease. The number of clones didn’t change after treatment, which suggests that pulsed dexamethasone therapy doesn’t affect the TCR repertoire.

When looked into the frequency of each clone, the data showed that most clones were of small size (<0.1%). These clones received little attention, because only when the TCR clones met with their corresponding antigens could they expand [26]. These small size clones might be naïve T cells (never met antigens) or memory T cells (their antigens had been removed), in other words, they were quiescent clones. On the other hand, the expanded clones might be the criminal of the disease, especially the HECs. In CD4^+^ T cells (after treatment) and CD8^+^ T cells (before and after treatment), HECs’ percentages were higher than other grades of clones. This was a hint of oligoclonal distribution, in accordance with former studies [38,39]. The oligoclonal TCR was also detected in RA by CDR3 sequencing method [27,36]. In the setting of RA, oligoclonal TCR was found in joint fluid only. The skewed distribution of TCR repertoire was not found in peripheral blood. It is tempting to speculate that oligoclonal proliferation is caused by stimulation of specific antigens in specific environment (joint fluid). In current study, the specific antigens, including GP IIb/IIIa, GPIb/IX, and GPV, existed in peripheral blood. Thus, the oligoclonal TCR detected in peripheral blood, could be stimulated by these antigens. As the antigens also existed in bone marrow, further sequencing the bone marrow derived T cells would help analyze the TCR CDR3 sequence.

We compared parameters before treatment with the counterparts after treatment. No significant change was found, except the percentage of grade 0.1%−0.2% clones in CD8^+^ T cells increased (from (3.14% ± 0.96%) to (3.93% ± 1.52%), *p* = 0.0431). A closer gaze at the clones showed that the number of clones increased from (22.60 ± 7.70) to (28.60 ± 11.26) after treatment (*p* = 0.0568), the percentage of each clone didn’t change after treatment. The underling mechanism of this change remains to be explored.

The fact that oligoclonal clones did not decrease after treatment was not in accordance with our assumption, nor in line with former reports. It was reported highly expanded T cell clones faded away after GVHD remitted [26,40,41]. In RA patients, the number and the percentage of HECs decreased after one year of treatment [27]. There are several presumable explanations: First, dexamethasone is not specifically targeting the HECs. Second, the disease remitted and platelet count increased in only one month. This short-term effect might be casted by inhibited B cells, as well as numerically and functionally recovered Treg cells [6,10]. Third, the life span of T cells differs a lot, one month period may be too short to see the T cell clones numerical changes, as well as the cytokines’ alterations. The follow up period in other researches focusing on the TCR repertoire were from several months to 1 year [26,27].

CDR3 sequence of HECs were found different from each other in ITP patients. This result is identical to the study in RA patients [27]. Another similar study in diabetes, the sequence of TCRα and TCRβ showed limited overlap (<5%) between TCRs in a specific antigen reactive CD4^+^ T cells [42]. It is known that the nucleotide sequence could be translated to amino acid sequence, however several different nucleotide sequence could be translated into a same amino acid sequence. Besides, study showed that nonamer peptides that bind to the same MHCII molecule only need to share five amino acids to cross-react on the same TCR [37]. So different CDR3 sequence could not be simply explained as the TCR clones recognize different antigens. The different CDR3 sequence between CD4^+^ T cells and CD8^+^ T cells, as these two kinds of TCR combined with different MHC molecules, didn’t necessarily mean they combined with different antigens for the same reason. However, our attempt to find autoreactive TCR did not seem to be gloomy. The tendency of VDJ usage showed that the immune repertoire of ITP was still of potential clinical value.

Chimeric antigen receptor (CAR)-engineered T cells (CAR-T cells) have yielded unprecedented efficacy in malignancies especially in B cell lymphoma, most remarkably in anti-CD19 CAR-T cells for B cell acute lymphoblastic leukemia (B-ALL) with up to a 90% complete remission rate. CAR-T therapy has made tremendous progress in recent years, as demonstrated by the remarkable clinical responses obtained from chimeric antigen receptor (CAR)-modified T cells (CAR-T) and T cell receptor (TCR)-engineered T cells (TCR-T). TCR-T uses specific TCRS optimized for tumor engagement and can recognize epitopes [43,44,45,46]. B cells in patients with autoimmune diseases can present their own antigens to autoimmune T cells to promote the release of inflammatory factors, or differentiate into plasma cells to release autoantibodies, which play an important role in the occurrence and development of autoimmune diseases. CAR T cells can kill B cells. CD19 CAR T cells had a significant and long-lasting effect in the treatment of systemic lupus erythematosus (SLE) in vivo trial [47]. This supports the application of CAR T cells in autoimmune diseases therapy. Although it is challenging, there are currently clinical trials of CAR T for the treatment of autoimmune diseases, such as SLE. Our data showed that CAR T cells is also a very promising treatment option for refractory ITP. By individualized designing novel CAR T based on the specificity of patients’ TCR, the safety and efficacy of CAR T treatment of ITP would be greatly improved.

The study has its limitations First, the sample size of the study, especially the NGS part, has been limited. Second, little attention was paid to the patients with platelet count between normal to over 30 × 10^9^/L. These patients resembled a mildly diseased group, whose T cells’ clonal alteration might be different. Third, although the study revealed individualized TCR CDR3 sequence of ITP, the underlying molecular mechanisms remained to be explored.

## 5. Conclusions

The present study demonstrated that traditional general study of serum cytokines and T cell subfamily could not comprehensively and accurately reflect the changes of T cell subsets. Sequenced TCR CDR3 inspecting the TCR repertoire of patients with ITP found that each of the ITP patients has a unique CD4^+^ or CD8^+^ T cells’CDR3 sequence, which not only highlights the importance of NGS CDR3 sequencing as an optimal modality to investigate the TCR repertoire, but also paves novel path for the researches of autoimmune diseases.

## Figures and Tables

**Figure 1 jcm-11-05665-f001:**
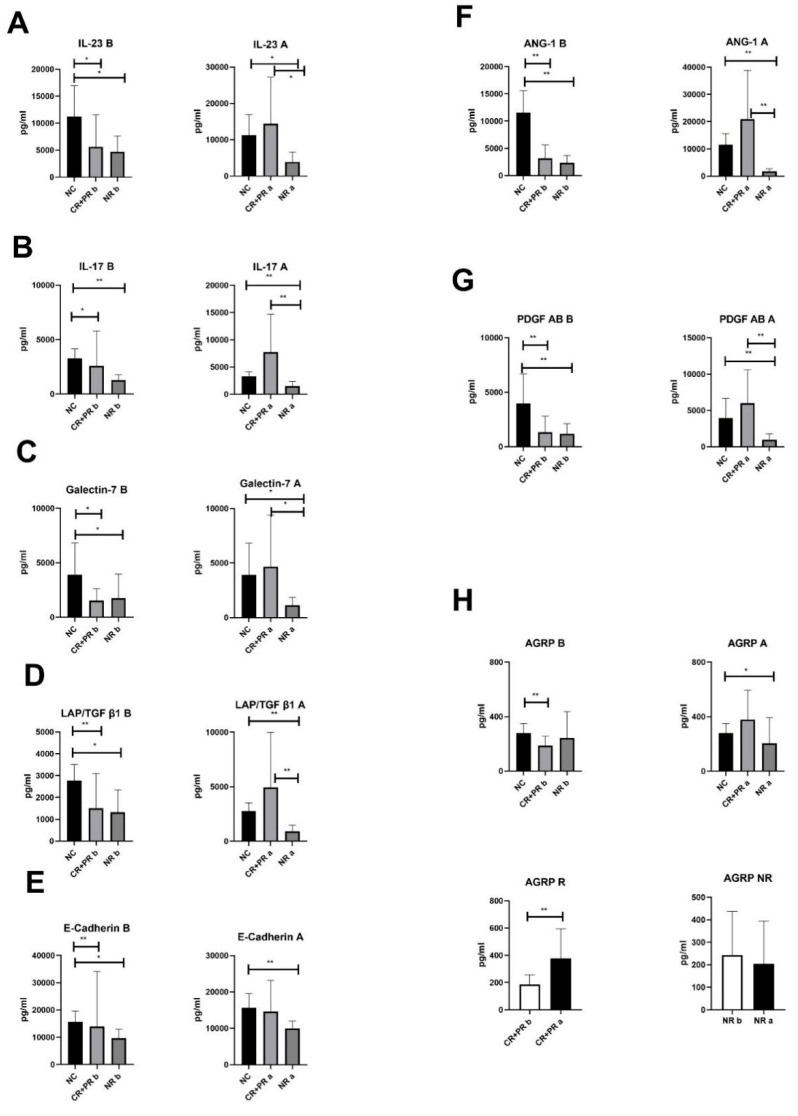
Changes of serum cytokines in ITP patients. (**A**) Changes of serum IL-23 in ITP patients; (**B**) Changes of serum IL-17β in ITP patients; (**C**). Changes of serum Galectin-7 in ITP patients; (**D**) Changes of serum LAP/TGF β1 in ITP patients; (**E**) Changes of serum PDGF-αβ1 in ITP patients; (**F**) Changes of serum ANG-1 in ITP patients; (**G**) Changes of serumE-cadherin in ITP patients; (**H**) Changes of serum AgRP in ITP patients; b: before treatment; a: after treatment; CR: complete remission; PR: partial remission; NR: no remission; * *p* < 0.05, ** *p* < 0.01. student *t* test and Wilcoxon rank-sum (Mann-Whitney) test were used for data fulfilled normal distribution and for those did not, respectively.

**Figure 2 jcm-11-05665-f002:**
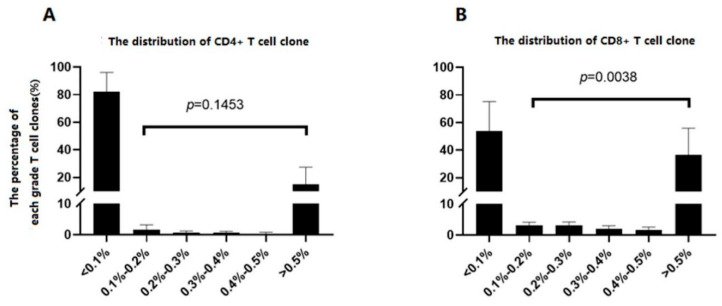
The clone distribution of ITP patients’ CD4^+^T and CD8^+^T cells before treatment. One Way ANOVA and Kruskal Wallis test were used for data fulfilled normal distribution and for those did not, respectively. (**A**) CD4^+^T cell clones; (**B**) CD8^+^T cell clones.

**Figure 3 jcm-11-05665-f003:**
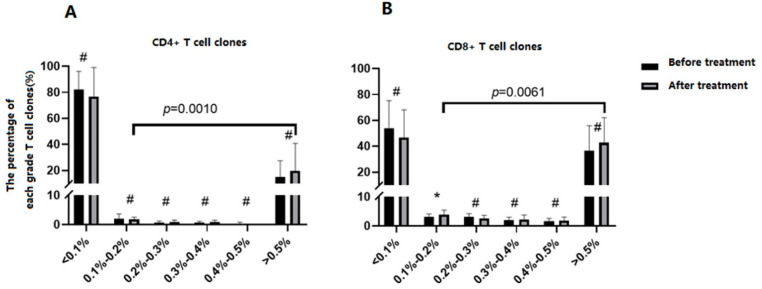
The clone distribution of ITP patients’ CD4^+^T and CD8^+^T cells after treatment, and the comparison with the distribution before treatment. One Way ANOVA and Kruskal Wallis test were used for data fulfilled normal distribution and for those did not, respectively. (**A**) CD4^+^T cell clones; (**B**) CD8^+^T cell clones. #: *p* > 0.05 when the percentage of T cell clones before treatment compared with the correspondent T cell clones after treatment; *: In the CD8^+^T cells, the number of clones of 0.1–0.2% grade increased after treatment, *p* = 0.0431.

**Figure 4 jcm-11-05665-f004:**
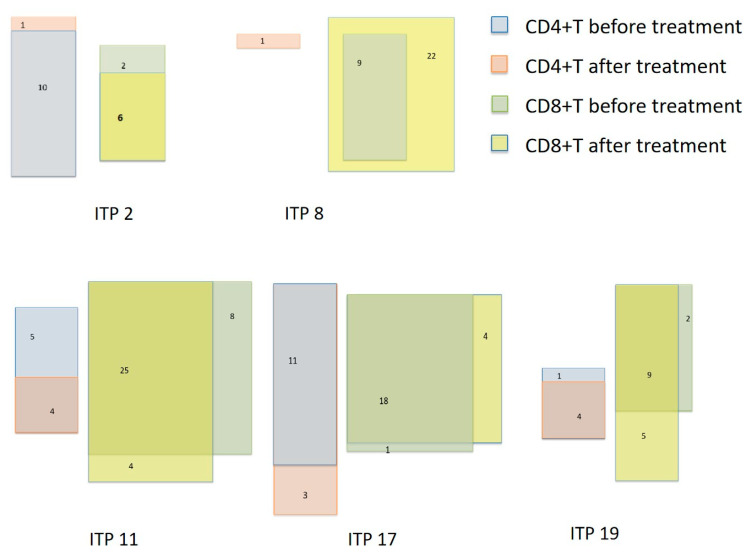
The diagram of CD4^+^T and CD8^+^T cells’ HECs comparison of patients. To clarify the diagram, ITP 2 was taken as an example: There were 10 CD4^+^ TCR HECs before treatment and eleven CD4^+^ TCR HECs after treatment. And the 10 CD4+ TCR HECs before treatment still existed as HECs after treatment. There were 8 CD8+ TCR HECs before treatment, and among them, 6 HECs existed after treatment. However, ITP 19 showed a little different alteration in CD8+ TCR HECs. Before treatment, there were 11 HECs. Among them, 9 HECs remained as HECs after treatment, while 2 HECs deminished. Another 5 new HECs emerged after treatment.

**Table 1 jcm-11-05665-t001:** The characteristic of enrolled patients.

Patient No.	Sex	Age (Years)	Platelet Counts (×10^9^/L)	
			Befor Treatment	After Treatment
1 *	F	28	3	210
2 #	F	19	5	101
3 *	M	29	11	118
4	F	49	4	28
5	F	33	22	102
6 *	M	71	18	99
7	F	22	5	157
8 #	F	45	25	116
9	F	76	2	11
10 *	M	29	6	55
11 #	F	58	15	102
12	M	64	28	27
13	F	68	17	26
14	M	36	23	204
15	M	34	8	104
16	F	67	6	15
17 #	F	49	18	123
18	F	39	23	27
19 #	F	62	15	139
20	M	47	12	92
21	M	38	11	155
22	F	55	21	180
23	F	57	6	17
24	M	18	15	134
25	F	77	14	198
26	M	66	5	95
27	F	27	17	24
	M:F = 10:17	47 (18–77)	14 (3–28)	102 (11–210)

* These 4 patients’ plasma samples were removed beacause of hemolysis in vitro. # These 5 patients’ DNA extracted from CD4^+^T and CD8^+^ T cells was adequate for later sequence analysis.

**Table 2 jcm-11-05665-t002:** The CD8^+^T cells’ TCR Vβ oligoclonal usage tendency testified by flow cytometry.

Vβ Subfamily	Before Treatment (n)	After Treatment (n)
Vβ 1	0	1
Vβ 2	6	4
Vβ 3	1	1
Vβ 5.1	15	18
Vβ 5.2	19	21
Vβ 5.3	7	6
Vβ 6.7	14	18
Vβ 7	7	9
Vβ 8	12	16
Vβ 9	16	13
Vβ 11	6	7
Vβ 12	14	13
Vβ 13.1	8	11
Vβ 13.6	8	8
Vβ 14	4	5
Vβ 16	2	5
Vβ 17	14	12
Vβ 18	7	8
Vβ 20	17	18
Vβ 21.3	10	18
Vβ 22	3	8
Vβ 23	3	0

**Table 3 jcm-11-05665-t003:** The distribution of CD4^+^ T cell repertoire in peripheral blood before treatment.

	Number of Clones <0.1%	Number of Clones 0.1–0.2%	Number of Clones 0.2–0.3%	Number of Clones 0.3–0.4%	Number of Clones 0.4–0.5%	Number of Clones >0.5%
ITP2	37,907	3	1	3	1	10
ITP8	541,495	1	0	1	0	0
ITP11	205,597	26	5	2	0	9
ITP17	218,117	17	5	3	1	11
ITP19	308,764	15	2	0	2	5

**Table 4 jcm-11-05665-t004:** The distribution of CD8+ T cell repertoire in peripheral blood before treatment.

	Number of Clones <0.1%	Number of Clones 0.1–0.2%	Number of Clones 0.2–0.3%	Number of Clones 0.3–0.4%	Number of Clones 0.4–0.5%	Number of Clones >0.5%
ITP2	285,477	15	7	2	0	8
ITP8	267,730	23	13	6	3	9
ITP11	239,437	25	19	9	6	33
ITP17	303,346	34	16	8	4	19
ITP19	128,323	16	9	3	5	11

**Table 5 jcm-11-05665-t005:** The distribution of CD4^+^ T cell repertoire in peripheral blood after treatment.

	Number of Clones <0.1%	Number of Clones 0.1–0.2%	Number of Clones 0.2–0.3%	Number of Clones 0.3–0.4%	Number of Clones 0.4–0.5%	Number of Clones >0.5%
ITP2	296,126	6	1	1	1	11
ITP8	301,674	13	4	1	0	1
ITP11	176,880	14	1	5	0	4
ITP17	253,060	19	7	3	0	14
ITP19	314,797	13	5	2	0	4

**Table 6 jcm-11-05665-t006:** The distribution of CD8+ T cell repertoire in peripheral blood after treatment.

	Number of Clones <0.1%	Number of Clones 0.1–0.2%	Number of Clones 0.2–0.3%	Number of Clones 0.3–0.4%	Number of Clones 0.4–0.5%	Number of Clones >0.5%
ITP2	439,425	15	6	1	1	6
ITP8	199,974	31	16	8	4	31
ITP11	195,202	37	13	9	8	29
ITP17	135,012	41	12	11	5	22
ITP19	172,172	19	6	3	3	14

**Table 7 jcm-11-05665-t007:** Representative sequence of CD8+ T cell CDR3 before and after treatment.

Sequence	Percentage before Treatment	The Percentage Rank of the Clone before Treatment	Percentage after Treatment	The Percentage Rank of the Clone after Treatment
TGTGCTGTGAGTGATCGGACTCTAGCAACACAGGCAAACTAATCTTT	0.200912	1	0.244993	1
TGTGCAATGAGAGAGAATAACTATGGTCAGAATTTTGTCTTT	0.042286	2	0.061844	2
TGTGCAGAGTGGGACGCAGGCAAATCAACCTTT	0.03282	3	0.018226	5
TGTGCTGTGAGCCCCATAATGCTGGCAACAACCGTAAGCTGATTTGG	0.024714	4	0.027579	3
TGTGCTGTGTTTAAGGGGCTCAAATTCCGGGTATGCACTCAACTTC	0.018299	5	0.026758	4
TGTGTGGTGAGCACTAACGACTACAAGCTCAGCTTT	0.017066	6	0.015805	6
TGTGCAATGAGAGAACGCAACAAATTTTACTTT	0.013396	7	0.015171	7
TGTGCTGTGGAAGACTATGGTCAGAATTTTGTCTTT	0.012513	8	0.0083	15
TGTGCTGGCCCCAAGCAAACCTCCTACGACAAGGTGATATTT	0.011513	9	0.008645	14
TGTGCTGTGATGGATAGCAACTATCAGTTAATCTGG	0.010112	10	0.0129	9
TGTGCTGTGAGAGATAGCAACTATCAGTTAATCTGG	0.010074	11	0.012218	10
TGTGCTGCCTTAATAATGCAGGCAACATGCTCACCTTT	0.009917	12	0.008811	13
TGTGTGGTGAGCGGGGAGGAGGAAACAAACTCACCTTT	0.008334	13	0.009348	12
TGTGCTGTGAGGGGTGCAGGCAACATGCTCACCTTT	0.008324	14	0.011192	11
TGTGCCCGAAACACCGGTAACCAGTTCTATTTT	0.007235	15	0.006974	16
TGTGCCTTTCGGTATGGAAACAAACTGGTCTTT	0.006343	16	0.004324	26
TGCATCCTGAGAGACTGTAGAGGCCAGACTCATGTTT	0.006316	17	0.006084	18
TGTGCTGTGCTGGACTCAGGAACCTACAAATACATCTTT	0.005898	18	0.006058	19
TGTGCTCTGAGCAACCCCCAAATTCAGGAAACACACCTCTTGTCTTT	0.005776	19	0.013743	8

## Data Availability

All data generated or analysed during this study are included in this published article.

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
