# Peer review of "TCR CDR3 Sequencing as a Clue to Elucidate the Landscape of Dysimmunity in Patients with Primary Immune Thrombocytopenia"

_jcm, 2022, doi:10.3390/jcm11195665_

Round 1

Reviewer 1 Report

This is an interesting study addressing the cytokine profiling and Tcell subsetsé TCR repertoire of patients with newly diagnosed ITP. However, the reevaluation of the patients' cell and cytokine has been performed hastily in the course of the disease. As the authors themselves mention in the text reevaluation has been performed much later in the case of other autoimmune diseases.

Results are contradictory with previous reports (eg IL17 levels in ITP flare). The authors need to provide an explanation for this. Maybe the timepoint of cytokine sampling in the course of the disease could account for this (Adams et al, Blood 2016)

Th1 cytokines were not addressed which however appear to be mainly affected in ITP and may be more strongly related to corticosteroid response. 

Finally, the manuscript needs language editing.

Author Response

Response to Reviewer 1

We sincerely thank you for the invaluable constructive comments and suggestions that have made our manuscript much stronger. After carefully reviewing the critiques and comments, the manuscript has been modified by authors accordingly. Your comments were listed in italic, followed by our response, indicating how the manuscript has been revised. We hope that the revised version of the manuscript is now acceptable for publication in the Journal of Clinical Medicine. Thank you very much for your time and help.

Reviewer 1:

Comments and Suggestions for Authors

  1. This is an interesting study addressing the cytokine profiling and Tcell subsetsé TCR repertoire of patients with newly diagnosed ITP. However, the reevaluation of the patients' cell and cytokine has been performed hastily in the course of the disease. As the authors themselves mention in the text reevaluation has been performed much later in the case of other autoimmune diseases.

Results are contradictory with previous reports (eg IL17 levels in ITP flare). The authors need to provide an explanation for this. Maybe the timepoint of cytokine sampling in the course of the disease could account for this (Adams et al, Blood 2016)

Response:We greatly appreciate your efforts giving us knowledgeable suggestions and offering us opportunity to improve the manuscript.

We totally agree with the esteemed review that the reason for cytokines results contradictory to previous report would be the relatively short time interval of cytokine sampling in the course of the disease. The patients’ serum cytokines were measured on enrollment and 1 month after initial treatment in this study. It should be better to follow the cytokines’ change for a longer time. The discussion section has been updated accordingly (Line 301-304 and Line 365). Thanks again.

  1. Th1 cytokines were not addressed which however appear to be mainly affected in ITP and may be more strongly related to corticosteroid response.

Response:We are grateful for your careful work and admirable suggestion. We agreed that Th1 cytokines are mainly affected in ITP, and are more strongly related to corticosteroid response. The Th1/Th2 imbalance was first described in 2004, some articles have discussed this topic. As the study focused on the T cell clones expanded in ITP patients’ peripheral blood, that each clone had an individualized TCR CDR3 sequence, Th1 cytokines profiling was not our priority.

  1. Finally, the manuscript needs language editing.

Response:Although English isn’t our first language, we have proof read the manuscript and corrected grammatical errors and typos, trying our best to make the whole manuscript readable and understandable. The manuscript has been read through by an English speaker.

Once again the authors are grateful for the knowledgeable reviewers time and help.

Reviewer 2 Report

Ji et al present a manuscript which has used TCR sequencing to identify autoreactive Tcells expansion in ITP patients. The manuscript does not bring novelty as other works have performed the same technique in patients with the same disease. Furthermore, the manuscript is poorly written, which compromises the understanding of the data. 

Author Response

Response to Reviewer 2

We sincerely thank you for the invaluable constructive comments and suggestions that have made our manuscript much stronger. After carefully reviewing the critiques and comments, the manuscript has been modified by authors accordingly. Your comments were listed in italic, followed by our response, indicating how the manuscript has been revised. We hope that the revised version of the manuscript is now acceptable for publication in the Journal of Clinical Medicine. Thank you very much for your time and help.

Reviewer 2:

Ji et al present a manuscript which has used TCR sequencing to identify autoreactive Tcells expansion in ITP patients. The manuscript does not bring novelty as other works have performed the same technique in patients with the same disease. Furthermore, the manuscript is poorly written, which compromises the understanding of the data.

Response: The authors are grateful for the learned reviewer’s input. In this study, NGS on TCR CDR3 was utilized to examine the oligoclonal T cells in ITP. The HECs alteration was also followed after treatment. The study exhibited a novel way to understand the dynamic situation of the TCR repertoire, thus paved a new path to identify the autoreactive T cell clones. To our best knowledge, this study is one of the first to report the identical HECs sequence in each individual patient with ITP, that has great potentials to improve individualized treatment for ITP patients, as well as other autoimmune diseases.

Although English isn’t our first language, the authors have tried their best to make the manuscript readable and understandable. The manuscript has been carefully modified according to the comments made by all esteemed reviewers. The language has been read through by an English speaker.

Once again the authors are grateful for the knowledgeable reviewers time and help.

Reviewer 3 Report

In this manuscript, the authors have shown with scientific rigor that CD4T and CD8T cells had over 2 million clones in ITP patients, before and after treatment, confirming the diversity of TCR repertoires. In addition they observed an individualized CD4+T 45 cell or CD8+T cell CDR3 sequence in each ITP patient. The study was conducted with a good scientifically sound. The research methodology is sufficiently described and conclusions are in accordance with the results. Perspectives are clearly formulated. An astute reader can easily determine the scientific scope of this research and understand its therapeutic potential.

Minor points :  

-       The authors should clarify several clinico-biological data of their healthy control subjects since this is the population to which the included patients are compared.

- The authors did not report any limitations in their study. In particular, there is a selection bias since the choice to include patients with platelet levels below 30,000 omitted ITP patients with higher platelet levels. Please elaborate on these two points. 

-       Figure 4 deserves more explanation for understanding the diagrams. An example could be given.

Author Response

Response to Reviewer 3

We sincerely thank you for the invaluable constructive comments and suggestions that have made our manuscript much stronger. After carefully reviewing the critiques and comments, the manuscript has been modified by authors accordingly. Your comments were listed in italic, followed by our response, indicating how the manuscript has been revised. We hope that the revised version of the manuscript is now acceptable for publication

Reviewer 3:

In this manuscript, the authors have shown with scientific rigor that CD4+ T and CD8+ T cells had over 2 million clones in ITP patients, before and after treatment, confirming the diversity of TCR repertoires. In addition they observed an individualized CD4+T 45 cell or CD8+T cell CDR3 sequence in each ITP patient. The study was conducted with a good scientifically sound. The research methodology is sufficiently described and conclusions are in accordance with the results. Perspectives are clearly formulated. An astute reader can easily determine the scientific scope of this research and understand its therapeutic potential.

Minor points: 

-     The authors should clarify several clinico-biological data of their healthy control subjects since this is the population to which the included patients are compared.

Response: Thanks very much for the esteemed reviews insightful comments. TCR repertoire studies have shown consolidate information in healthy population. Related references have been cited in the revised manuscript (references 24, 25).The study focused on the sequence of TCR CDR3, the differences between CD4+T and CD8+T cells and the differences between before and after treatment were compared, NGS was not performed in healthy control subjects.

- The authors did not report any limitations in their study. In particular, there is a selection bias since the choice to include patients with platelet levels below 30,000 omitted ITP patients with higher platelet levels. Please elaborate on these two points.

Response: Thank you very much for your valuable suggestion. Study limitations have been acknowledged in discussion section (Line 402-407).

-    Figure 4 deserves more explanation for understanding the diagrams. An example could be given.

Response: Thanks very much for the advice. Per your suggestion, explanation and example have been added in Figure 4.

Once again the authors are grateful for the knowledgeable reviewers time and help.

Round 2

Reviewer 1 Report

The points raised in the previous review have been adequately addressed by the authors.

Author Response

Thank you very much! Very grateful.

Reviewer 2 Report

In the figure legends, specify the statistical test utilized.

Author Response

Thank you very much for your help. We addressed the statistical test utilized in each figure legends.